# Intestinal Epithelial AMPK Deficiency Causes Delayed Colonic Epithelial Repair in DSS-Induced Colitis

**DOI:** 10.3390/cells11040590

**Published:** 2022-02-09

**Authors:** Séverine Olivier, Hanna Diounou, Camille Pochard, Lisa Frechin, Emilie Durieu, Marc Foretz, Michel Neunlist, Malvyne Rolli-Derkinderen, Benoit Viollet

**Affiliations:** 1Université de Paris, Institut Cochin, CNRS, INSERM, F-75014 Paris, France; sev-olivier@orange.fr (S.O.); hanna.diounou@inserm.fr (H.D.); lisa.frechin@orange.fr (L.F.); marc.foretz@inserm.fr (M.F.); 2Université de Nantes, TENS, The Enteric Nervous System in Gut and Brain Disorders, Institut des Maladies de l’Appareil Digestif, F-44093 Nantes, France; camille.pochard@etu.univ-nantes.fr (C.P.); emilie.durieu@univ-nantes.fr (E.D.); michel.neunlist@univ-nantes.fr (M.N.); malvyne.derkinderen@univ-nantes.fr (M.R.-D.)

**Keywords:** AMPK, metformin, inflammatory bowel disease, intestinal barrier integrity, epithelium repair, goblet cells, pro-inflammatory cytokines

## Abstract

Dysfunctions in the intestinal barrier, associated with an altered paracellular pathway, are commonly observed in inflammatory bowel disease (IBD). The AMP-activated protein kinase (AMPK), principally known as a cellular energy sensor, has also been shown to play a key role in the stabilization and assembly of tight junctions. Here, we aimed to investigate the contribution of intestinal epithelial AMPK to the initiation, progression and resolution of acute colitis. We also tested the hypothesis that protection mediated by metformin administration on intestinal epithelium damage required AMPK activation. A dextran sodium sulfate (DSS)-induced colitis model was used to assess disease progression in WT and intestinal epithelial cell (IEC)-specific AMPK KO mice. Barrier integrity was analyzed by measuring paracellular permeability following dextran-4kDa gavage and pro-inflammatory cytokines and tight junction protein expression. The deletion of intestinal epithelial AMPK delayed intestinal injury repair after DSS exposure and was associated with a slower re-epithelization of the intestinal mucosa coupled with severe ulceration and inflammation, and altered barrier function. Following intestinal injury, IEC AMPK KO mice displayed a lower goblet cell counts with concomitant decreased Muc2 gene expression, unveiling an impaired restitution of goblet cells and contribution to wound healing process. Metformin administration during the recovery phase attenuated the severity of DSS-induced colitis through improvement in intestinal repair capacity in both WT and IEC AMPK KO mice. Taken together, these findings demonstrate a critical role for IEC-expressed AMPK in regulating mucosal repair and epithelial regenerative capacity following acute colonic injury. Our studies further underscore the therapeutic potential of metformin to support repair of the injured intestinal epithelium, but this effect is conferred independently of intestinal epithelial AMPK.

## 1. Introduction

The intestinal barrier is the greatest surface of the body in contact with the external environment, allowing the transfer of nutrients to the organism, but it is also essential to maintain an efficient protection against toxins and bacteria. This intestinal barrier arises from the diversity of its architecture and constituents; it includes the mucus layer (with secreted immunoglobulins and antimicrobial peptides), *lamina propria* (containing lymphoid tissues with macrophages and peyers patches), mesenteric lymph nodes (hosting immune cells) and the intestinal epithelium, also called the intestinal epithelial barrier (IEB). Cells that constitute this epithelium are organized in a monolayer containing enterocytes, goblet cells, paneth cells, enteroendocrine cells, M cells and Tuft cells, as well as intestinal stem cells [1]. Dysfunctions of the IEB function are associated with increased intestinal permeability or “leaky gut” in the pathogenesis of several diseases, including both intestinal (inflammatory bowel diseases (IBD), such as Crohn’s disease and ulcerative colitis) and extra-intestinal (as reported in obesity, diabetes and nonalcoholic fatty liver disease) disorders [1,2]. Thus, interventions strengthening the IEB function could be a promising therapeutic option to improve gut epithelial health.

AMP-activated protein kinase (AMPK) is an evolutionary-conserved nutrient-sensitive protein kinase that regulates cellular metabolism to maintain energy homeostasis. AMPK is a heterotrimeric complex composed of one catalytic α subunit, and two regulatory β and γ subunits, which occur as multiple isoforms (α1/α2; β1/β2; γ1/γ2/γ3) encoded by distinct genes [3]. Under low energy conditions (characterized by rising AMP:ATP ratio), AMPK is activated by the upstream liver kinase B1 (LKB1) that phosphorylates the catalytic AMPKα subunit on Thr-172 residue. Activated AMPK phosphorylates many downstream targets to switch on ATP-producing processes while simultaneously switching off energy-consuming processes, thus acting to restore energy homeostasis. In addition, in response to elevation of cytoplasmic Ca^2+^ levels, the calcium/calmodulin-dependent protein kinase kinase 2 (CaMKK2) activates AMPK by phosphorylating the catalytic AMPKα subunit on Thr-172 residue, providing a Ca^2+^-activated pathway to switch on AMPK independently of changes in the AMP:ATP ratio and LKB1 phosphorylation. Recent studies have highlighted the role of AMPK, independently of its function as a metabolic sensor [3], in the regulation of epithelial tight junction (TJ) assembly and maintenance of epithelial barrier function [4,5,6]. Considering the importance of IEB in gut homeostasis, alleviating intestinal disorders by activating AMPK appears to be an attractive therapeutic strategy. Several studies have addressed the role of AMPK signaling in the protection of intestinal mucosal barrier function mediated by the supplementation of specific nutritional compounds in in vitro models of cultured intestinal epithelial cells [7,8]. In addition, butyrate, a microbial metabolite, has been shown to increase AMPK activity and promote the development of barrier function by accelerating tight junctions (TJs) assembly in Caco2 cells [9]. It has been also reported that pharmacological AMPK activation by 5-aminoimidazole-4-carboxamide-1-β-D-ribofuranoside (AICAR) is capable to enhance barrier function and epithelial differentiation in vitro [10,11]. Consistently, we recently showed that pharmacological activation of AMPK by the direct AMPK activator 991 protects the TJs from disassembly induced by calcium depletion and ensures a better recovery of epithelial barrier function in Caco2 cells [12]. However, the role of AMPK in IEB function and maintenance under pathological conditions is still relatively less explored in vivo.

In line with improved barrier function, various studies have shown the beneficial effects of the anti-diabetic compound metformin, acting through AMPK-dependent and -independent mechanisms [13,14], on dextran sulfate sodium (DSS)-induced colitis in mice [15,16,17,18]. Metformin has been reported to act on intestinal inflammation by inhibiting the NF-κB pathway [18] and attenuating the production of the pro-inflammatory cytokine IL-6, which plays a key role in amplifying inflammatory signals in the gut [17]. These effects of metformin on inflammation and IEB dysfunction appears to occur in an AMPK-dependent manner [16]. However, our knowledge on the exact contribution of intestinal epithelial AMPK in the therapeutic action of metformin remains elusive.

While accumulating evidence has shown the physiological importance of AMPK in regulating IEB function under homeostatic conditions [11,19], first, we questioned the role of IEC AMPK in the pathogenesis of colitis by using the DSS-induced experimental colitis model, in which acute intestinal inflammation with epithelial loss is induced by treatment with DSS [20]. Second, we investigated the contribution of IEC AMPK to the beneficial effect of metformin in the treatment of colitis. The identification of new targets for therapy may be relevant for future applications for IBD patients.

## 2. Materials and Methods

### 2.1. Reagents and Antibodies

Dextran sodium sulfate (DSS; molecular weight 36–50 kDa, #160110) was purchased from MP Biomedicals. TRITC-conjugated dextran (4 kDa) was obtained from TdB Consultancy AB. Primary antibodies directed against total AMPKα (#2532), AMPKα phosphorylated at Thr-172 (#2531) and β-actin (#4967) were purchased from Cell Signaling Technology. Ki67 antibody (ab15580) was purchased from Abcam. Circulating cytokines were analyzed using the U-PLEX cytokine assay from Meso Scale Diagnostics.

### 2.2. Mice

All animal procedures were carried out in accordance with the EU guidelines for the protection of vertebra animals used for scientific purposes (2010/63/EU) and were approved by the Institutional Animal Care and Use Committee n°034 from Université de Paris (APAFIS 14911-2017120813494332). Male and female mice, eight to twenty-two weeks old, with a C57Bl/6 genetic background were used for all experiments. Mice were bred hemizygous for the Cre allele, resulting in AMPK α1^lox/lox^/α2^lox/lox^-villin-CRE^-^ and -CRE^+^ littermates. AMPK α1^lox/lox^/α2^lox/lox^-villin-CRE^+^ mice were constitutively deleted for the two AMPKα1/α2 catalytic subunits, specifically in IEC [8]. AMPKα1^lox/lox^/α2^lox/lox^ animals served as control. All mice were housed in conventional animal facilities under controlled environment conditions (12-h light/dark cycle and temperature maintained at 20 °C) with free access to acidified water and standard mouse diet. Acid-free drinking water was given to mice during all experimental procedures. To limit variability between cages, age- and sex-matched littermates of both genotypes were kept co-housed throughout all the experiments.

### 2.3. DSS-Induced Acute Colitis

Acute colitis was induced in female mice by adding 4% (*w*/*v*) DSS to the acid-free drinking water for 4 days. Regenerative response of the intestinal epithelium to DSS treatment was studied in female mice exposed to 2.5% (*w*/*v*) DSS in drinking water for 5 days, followed by an additional 6 days of recovery with regular water. Animal body weights, stool consistency and the presence of gross blood in feces were recorded daily throughout all the experiments. Animals were sacrificed at the end of the treatment by cervical dislocation. Colonic tissues were both stored at −80 °C for further gene expression analysis and were also fixed for 24 h in 10% (*v*/*v*) neutral-buffered formalin solutions for complementary histopathological analysis.

### 2.4. Histopathological Analysis

After cervical dislocation, intestine was collected and Swiss rolls of colon were fixed in 4% buffered formalin for 24 h, dehydrated and embedded in paraffin, and 4 µm thick sections were stained with Hematoxylin/Eosin. Colonic histological damage was scored in a blinded manner by quantifying destruction of mucosal architecture, cellular infiltration, muscle thickening and loss of goblet cells. The extent of destruction of normal mucosal architecture was scored as 0–3 (0 = no destruction, 1 = 1/3 basal destruction, 2 = 2/3 basal destruction, 3 = loss of crypt and epithelium). The presence and degree of cellular infiltration was also scored as 0–3 when the infiltration was normal, around the crypt basis, reaching the muscularis mucosae and reaching the submucosa, respectively. The extent of muscle thickening was scored as 0–3 when the thickening was none, mild, moderate or massive, respectively. The presence or absence of goblet cell depletion was scored as 0 (normal) or 1 (massive depletion). An extension factor of 1–4 was applied when the criteria measured reached 25%, 50%, 75%, or 100% of the fragment analyzed. The abundance of goblet cells (mucin) was evaluated in alcian blue-stained colon sections according to standard histology protocols. Cell proliferation was evaluated by counting Ki67 positive cell over total crypt length. Ki67 staining was performed by using anti-Ki67 antibody using the standardized streptavidin-biotin detection system HRP-DAB method.

### 2.5. In Vivo Intestinal Permeability Assay

At indicated, intestinal permeability was assessed by oral gavage of mice with 500 mg/kg body weight TRITC-dextran (4 kDa) dissolved in saline containing 0.5% carboxymethylcellulose. Blood was collected at the tail tip 4 h after gavage and plasma was used to determine TRITC fluorescence intensity. Standard curves were used to evaluate TRITC (excitation: 544 ± 10 nm; emission: 580 ± 10 nm) concentrations and fluorescent intensity in plasma of each sample was measured using the microplate reader CLARIOstar^®^ Plus (BMG Labtech, Ortenberg, Germany).

### 2.6. Quantification of Lipocalin-2

After feces collection on the indicated days, samples were diluted and extracted in cold PBS-antiprotease (PBS containing Complete Protease Inhibitor Cocktail; Roche^®^). Protein concentration was determined by a spectrophotometric method with a BCA kit (ThermoScientific, Waltham, MA, USA), following the manufacturer’s instructions. Concentrations of lcn-2 in feces were determined by using the DuoSet^®^ ELISA Development System (R&D Systems, Minneapolis, MN USA) for mouse Lipocalin-2 (DY1857). Briefly, after coating a microplate high absorbance with capture antibody, the unspecific binding sites were blocked. Standards and samples were diluted (according to previously optimized dilutions) and added to the plate, which was incubated 2 h. After washing, the enzyme-linked detection antibody was added to each well and left again for 2 h. A working dilution of Streptavidin-HRP was added for 20 min, and the substrate solution (H_2_O_2_ and tetramethylbenzidine) was used to visualize the enzymatic reaction, which was stopped by adding sulfuric acid. The optical density of each well was determined with a spectrophotometer (BioTek^®^, Winooski, VT, USA) at 450 nm.

### 2.7. Cell Culture and Measure of TEER

AMPK KO and control WT Caco2 cells were described previously [12,21]. Cells were maintained at 37 °C with 5% CO_2_ and 95% air atmosphere. Cells were grown in MEM (Life Technologies, Carlsbad, CA, USA) supplemented with 20% FBS, 1% penicillin/streptomycin and 1% non-essential amino acids (Life Technologies, Carlsbad, CA, USA). To study wound healing of differentiated and polarized Caco-2 cells, 135,000 AMPK KO and WT Caco-2 cells were seeded onto twelve-well Transwell filters (0.40 μm porosity, Corning, NY, USA) and cultured for 15 days after reaching confluence. Barrier integrity was monitored over time after wounding by measurements of transepithelial electrical resistance (TEER), obtained in ohm square centimeters, using the cellZscope online monitoring device (Proteigene, Saint-Marcel, France).

### 2.8. Cell Adhesion Assay and xCELLigence Real-Time Cell Analysis

Adhesion was quantified through plating 2000 WT or AMPK KO Caco2 cells per well for 15, 30, 60, 120 or 180 min. After the indicated time, non-adherent cells were washed out using PBS. The remaining adherent cells were fixed with PAF 4% for 30 min before staining with DAPI. The cells were counted from 4 pictures (4× magnification) per well (two wells per condition). To assess cell growth, we used the xCELLigence label-free real-time cell analysis platform. Briefly, cells were seeded at 14,000 cells/well in designated 96-well plates and were maintained at 37 °C with 5% CO_2_ according to manufacturer’s instructions. Cell growth was dynamically monitored every 15 min for 72 h using the real time cell assay (RTCA) software by calculating cell index, a dimensionless parameter which is directly correlated to the proportion of the plate surface occupied by adherent cells. Proliferation rate is determined by calculating the slope of the line between two time points during the assay.

### 2.9. Scratch Wound Assay

To investigate the wound healing effect on Caco-2 cells, the IncuCyte instrument and technology was used (Essen BioScience, Ann Arbor, MI, USA). The cells were seeded (in triplicates) in 96-well plate and cultured in DMEM L-Glutamax culture medium plus 20% FBS under a humidified atmosphere of 5% CO_2_ and 95% air at 37 °C until a confluent monolayer of cells was formed. Uniform scratches were implemented on the cell-monolayer surface using the WoundMaker (Essen BioScience, Ann Arbor, MI, USA) device to obtain homogeneous wounds (700–800 μm width scratches devoid of cells), and rinsed in PBS to remove detached cells and debris. Each wound image was automatically recorded by IncuCyte optical module phase contrast with a 10× objective, every 3 h for 60 h. Digital images were analyzed using IncuCyte software (version 2020B; Essen BioScience, Ann Arbor, MI, USA) to quantify cell motility expressed as a relative percentage of the changes overtime in the distance between wound boundaries. To study wound healing of differentiated and polarized Caco-2 cells, AMPK KO and WT Caco-2 cells cultured for 15 days after reaching confluence were wounded by using a tip attached to a 0.5- to 10-μL pipette. Each wound was photographed at 0 and 48 h by using a microscope (Axio Observer; Zeiss, Jena, Germany).

### 2.10. Western Blotting

Protein extracts were obtained by homogenizing full-thickness colon biopsies in ice-cold lysis buffer (50 mM Tris, pH 7.4, 1% Triton X-100, 150 mM NaCl, 1 mM EDTA, 1 mM EGTA, 10% glycerol, 50 mM NaF, 5 mM sodium pyrophosphate, 1 mM Na_3_VO_4_, 25 mM sodium-β-glycerophosphate, 1 mM DTT, 0.5 mM PMSF and protease inhibitors cocktail (Complete Protease Inhibitor Cocktail; Roche, Basel, Switzerland) by the use of a TissueLyzer (Qiagen, Hilden, Germany). Homogenates were sonicated on ice for 15 s to shear DNA and reduce viscosity and were centrifuged at 10,000× *g* for 10 min at 4 °C. The supernatants were collected for the determination of the protein concentration with a BCA protein assay kit (Thermo Fisher Scientific, Waltham, MA, USA). Protein samples were dissolved in water and Laemmli loading buffer (Tris 0.75 M pH 6.8; 40% glycerol; 20% SDS; 10% β-mercaptoethanol, 2% bromophenol blue) at 1 µg/µL. Proteins (25 μg) were separated by SDS-PAGE in precast 4–15% polyacrylamide gels (Biorad, Hercules, CA, USA) and the resulting bands were transferred to nitrocellulose membranes. Equal loading was checked by membrane staining with Ponceau Red before blocking with Tris-buffered saline supplemented with 0.2% NP40 and 5% non-fat dry milk for 30 min at room temperature. Immunoblotting was performed following standard procedures. Total pan-AMPKα and phosphorylated AMPKα Thr172 were probed from separated membranes. The signals were detected with chemiluminescence reagents (EMD Millipore, Billerica, MA, USA) by using ImageQuant LAS 4000 control software version 1.2 (GE Healthcare, Chicago, IL, USA).

### 2.11. Quantitative Real-Time PCR Analysis

Fragments of the different intestinal segments were lysed in Trizol-reagent (Sigma Aldrich, St. Louis, MO, USA) and total RNA extraction was performed with the Nucleospin RNAII kit according to the manufacturer’s recommendations (Machery-Nagel, Düren, Germany). One μg of purified mRNA was denatured and retro-transcribed using Superscript III reverse transcriptase (Invitrogen, Waltham, MA, USA). PCR amplifications were performed using the Absolute Blue SYBR green fluorescein kit (Roche, Basel, Switzerland) and analyzed on the StepOnePlus system (Life Technologies, Carlsbad, CA, USA). Relative gene expression was calculated using the comparative Ct (2^−∆∆Ct^) method, where values were normalized to a housekeeping gene. The following primers were used: Tumor necrosis factor α (TNFα; Forward Primer: 5′-GAA CTT CGG GGT GAT CGG TCC-3′, Reverse Primer: 5′-GCC ACT CCA GCT GCT CCT CC-3′), Interleukin 1β (IL-1β; Forward Primer: 5′-GCC TCG TGC TGT CGG ACC CAT A-3′, Reverse Primer: 5′-TTG AGG CCC AAG GCC ACA GGT-3′), Interleukin 6 (IL-6; Forward Primer: 5′-TCC AGT TGC CTT CTT GGG AC-3′, Reverse Primer: 5′-AGT CTC CTC TCC GGA CTT GT-3′), Zona occludens-1 (ZO-1; Forward Primer: 5′-AAG AAT ATG GTC TTC GAT TGG-3′, Reverse Primer: 5′-ATT TTC TGT CAC AGT ACC ATT TAT CTT C-3′), Occludin (Forward Primer: 5′-ATG TCC GGC CGA TGC TCT C-3′, Reverse Primer: 5′-TTT GGC TGC TCT TGG GTC TGT AT-3′), Mucin-2 (Muc-2; Forward Primer: 5′-CCC AGA AGG GAC TGT GTA TG-3′, Reverse Primer: 5′-TGC AGA CAC ACT GCT CAC A-3′) and Ribosomal protein S6 (RPS6; Forward Primer: 5′-CCA AGC TTA TTC AGC GTC TTG TTA CTC C-3′, Reverse Primer: 5′-CCC TCG AGT CCT TCA TTC TCT TGG C-3′).

### 2.12. Statistical Analysis

Data are presented as means ± SD. The results were analyzed using Student’s *t* test, non-parametric Mann–Whitney test, one-way ANOVA or two-way ANOVA, as appropriate, followed by a Bonferroni post hoc test using the GraphPad Prism software. A value of *p* < 0.05 was considered statistically significant.

## 3. Results

### 3.1. IEC-Specific AMPK α1/α2 Deficiency Influenced Intestinal Inflammation upon DSS-Induced Epithelial Injury

To explore the clinical importance of AMPK in IECs and its contribution in the control of mucosal barrier integrity, we utilized an inflammatory injury model in which colitis is induced by administration of 4% (*w*/*v*) DSS in drinking water for 4 days to promote intestinal inflammation (Figure 1A). There were no noticeable differences in body weight changes and colon lengths, a widely used measure of injury in DSS colitis, between IEC AMPK KO and WT mice after DSS challenge (Figure 1B,C). The severity of the tissue damage and inflammation was also not different between IEC AMPK KO and WT mice as evidenced by similar increase in in vivo paracellular permeability (4 kDa dextran flux) (Figure 1D). However, although enhanced expression of thepro-inflammatory gene IL-6 was similar on day 4 of 4% DSS-induced colitis measured by real-time quantitative PCR in full-thickness samples taken from the colons of WT and IEC AMPK KO mice, there was a significant increase in proinflammatory cytokines TNFα and IL-1β gene expression levels in DSS-treated IEC AMPK KO mice, indicating higher colonic inflammation in the absence of IEC AMPK in response to acute experimental ulcerative colitis (Figure 1E).

### 3.2. Loss of IEC AMPK α1/α2 Caused Impaired Recovery from DSS-Induced Epithelial Injury

To evaluate the impact of AMPK deletion on colonic mucosal injury and epithelial wound repair after colitis, IEC AMPK KO and WT mice were exposed to 2.5% (*w/v*) DSS in drinking water for 5 days followed by an additional 6 days of recovery with regular water (Figure 2A). The survival rate was similar in IEC AMPK KO and WT mice at the terminal endpoint (Figure 2B). DSS-induced colitis was successfully and equally induced in both IEC AMPK KO and WT mice during the acute phase of the disease on day 7, as characterized by similar body weight loss and the appearance of loose feces and significant diarrhea with visible rectal bleeding during disease progression (Figure 2C). Clinical symptoms were accompanied by a shortening of the colon on day 10 in both WT and IEC AMPK KO mice (Figure 2D). However, at the end of the recovery phase, weight loss differed significantly between the two genotypes. WT mice exhibited accelerated body weight gain and gradually recover faster than IEC AMPK KO mice (Figure 2C). In accordance with a delay in the restoration of intestinal barrier integrity, IEC AMPK KO mice displayed a higher paracellular intestinal permeability on day 9 (4 days after DSS removal) as compared to WT mice (Figure 2E). In addition, expression levels of the tight junction protein occludin was also lower in IEC AMPK KO compared to WT mice (Figure 2F). Histologic examination of Swiss-rolled colons in the colitis recovery period demonstrated a typical histological architecture in WT mice with predominant distal injury characterized by epithelial ulceration and inflammatory infiltrates, whereas IEC AMPK KO mice displayed a larger area of epithelial erosion with more pronounced mucosal ulceration, and more massive inflammatory cell infiltration (Figure 2G–I). In line with this, expression of the pro-inflammatory cytokine IL-1β was significantly higher in full-thickness biopsies taken from distal colons of IEC AMPK KO compared to WT mice on day 9 of the recovery phase (Figure 2J). In addition, plasma IL-1β levels were also significantly increased in IEC AMPK KO mice (Figure 2K), indicating that loss of IEC AMPK activity might contribute to the development of a low grade basal chronic inflammation during DSS-induced colitis. Altogether, these findings indicate that the absence of AMPKα1 and AMPKα2 in the intestinal epithelium leads to increased sensitivity to DSS-induced colitis associated with a delay in mucosal repair.

### 3.3. AMPK Signaling Supports Epithelial Proliferation Following DSS-Mediated Injury

To determine if mucosal epithelial loss in DSS-treated IEC AMPK KO mice is primarily due to impaired intestinal stem cell (ISC) regeneration, we examined the sensitivity of IEC AMPK KO and WT mice to irradiation-induced gut injury. In irradiated IEC AMPK KO mice, histological analysis of colon sections showed that crypt length and the number of dead and regenerative crypts were significantly increased compared to untreated animals, as observed in WT mice (Figure 3A,B). These data indicate that IEC AMPK deficiency does not contribute to the regenerative capacity of the ISCs, particularly during recovery from injury, but could interplay later in cell proliferation and/or differentiation. Next, to further characterize the changes that underlie the delayed mucosal repair in IEC AMPK-deficient mice after DSS administration, we examined the proliferative capacity of the intestinal epithelium. Proliferation of epithelial cells during the course of DSS-induced colitis was assessed in IEC AMPK KO and WT mice by immunostaining for Ki67, a nuclear marker of cell proliferation. We found a marked attenuation of the proliferative response to DSS-induced injury on day 8 following the injury phase in IEC AMPK KO mice compared to WT mice (Figure 3C), indicating an alteration in the regenerative process in the absence of AMPK. However, during the recovery period, while the percentage of Ki67-positive cells in the distal colon from WT mice reached a plateau, it was significantly enhanced in IEC AMPK KO mice on days 10 and 11 (Figure 3C,D). Ki67 accumulation reflects the presence of actively proliferating cells and suggests accelerated cell cycle progression within the proliferating crypt compartment in the intestinal epithelium of IEC AMPK KO mice [22]. This stimulated proliferation in crypts of epithelial AMPK-deficient colon could be due to immune cells and/or mesenchymal cells underneath crypts, which secrete growth factors and cytokines crucial for crypt cell proliferation and tissue regeneration. Next, to address the role of AMPK signaling on IEC proliferation, we performed xCELLigence real-time cell analysis on Caco2 cells lacking both AMPKα1 and AMPKα2 subunits [12]. AMPK-deficient (AMPK KO) Caco2 cells showed a lower cell index value and proliferation rate compared to WT Caco2 cells (Figure 3E,F). However, the adhesion of WT and AMPK KO Caco2 cells was not different (Figure 3G), excluding this process in the difference observed. These data suggest a potential role for AMPK in the control of IEC proliferation in vivo.

### 3.4. Absence of AMPK Is Not Detrimental for Cell Spreading Response in Caco2 Cells but Necessary for Mature Barrier Establishment

As mucosal wound repair in the intestine requires coordinated epithelial proliferation, migration and differentiation, we next investigated the role of AMPK during epithelial wound healing using two models. First, we introduced a scratch on a cell monolayer from control and AMPK-deficient Caco2 cells platted on plastic wells and monitored wound closing by time-lapse microscopy. We observed that upon mechanical damage, wound closure occurred at similar rate between WT and AMPK-deficient cells (Figure 4A,B). Second, we also performed wound healing onto monolayers of differentiated and polarized AMPK KO and WT Caco2 cells grown on Transwell filters and followed long-term monolayer integrity by measuring recovery of transepithelial electrical resistance (TEER) during and after wound closure. Two days after wounding, while the wound is visually closed (Figure 4C), the TEER of AMPK-deficient Caco2 cells was lower compared to control cells, suggesting that spreading was not affected by AMPK deficiency, but the capacity to re-establish the epithelial barrier function was strongly altered (Figure 4D).

### 3.5. Impaired Restitution of Mucus-Producing Goblet Cells in the Colon of IEC AMPK KO Mice Following DSS-Induced Colitis

Healing of the gastrointestinal mucosa after injury is a multistep process that requires restoration of a continuous epithelial monolayer in concert with the restitution of goblet cells to accomplish full repair and function of the mucosa. Restitution of goblet cells during the recovery phase of DSS-induced colitis was addressed by staining colon section from IEC AMPK KO and WT mice with Alcian blue (Figure 5A). On day 10, IEC AMPK KO mice displayed fewer mucus-producing goblet cells than WT mice, corroborating that the differentiation of the hyperproliferative crypts is delayed in the absence of AMPK (Figure 5B). The decreased quantity of goblet cells was associated with a strong reduction of the goblet cell marker mucin (Muc)-2 expression in the colon of IEC AMPK KO as compared to WT mice following DSS challenge on day 10 (Figure 5C). Overall, our data demonstrate that absence of IEC AMPK during colitis induction impairs the goblet cell number, with concomitant decreased Muc2 expression.

### 3.6. Metformin Ameliorated DSS-Induced Colitis in an IEC AMPK-Independent Manner

Next, to examine the consequence of AMPK activation after the induction of DSS-induced colitis on mucosal repair, IEC AMPK KO and WT mice received metformin in drinking water at the beginning of the recovery phase after withdrawal of DSS (Figure 6A). IEC AMPK KO and WT mice treated with a dose of metformin at 2 mg/mL for 4 days showed no relevant effects on body weight changes (Figure 6B), improvement of the intestinal epithelial barrier function on day 9 (Figure 6C), nor reduction in intestinal inflammation, as revealed by the measure of fecal Lcn-2 levels (Figure 6D). Subsequent groups of mice were then administrated with a higher dose of metformin at 10 mg/mL for 5 days (Figure 6E). Metformin administration was associated with increased AMPK Thr172 phosphorylation in the colon of WT but not IEC AMPK KO mice (Figure 6F,G). No relevant effects were noticed on body weight regain during the recovery phase in metformin-treated IEC AMPK KO and WT mice (Figure 6H). In addition, high dosage of metformin was not effective to significantly improve gut leakiness in WT nor in IEC AMPK KO mice following DSS challenge (Figure 6I). However, we observed that metformin administration substantially enhanced re-epithelization of intestinal epithelium in both WT and IEC AMPK KO mice, as illustrated by a significant amelioration in histological score and reduction of colon ulceration after damage (Figure 6J,K). Collectively, these data suggest that metformin administration during the recovery phase ensures adequate epithelial repair through an IEC AMPK-independent mechanism.

## 4. Discussion

IBD is a major public health issue associated with high morbidity and mortality, and studying the underlying mechanisms is important for the development of new therapeutic strategies [23]. The loss of IEB integrity and inefficient mucosal repair are important factors triggering IBD onset [1,24]. It has been reported that IBD is clearly associated with IEB dysfunction (leaky gut) in patients, and barrier abnormalities include, among others, intestinal epithelial cell–cell junction alterations [25]. The 36–50 kDa DSS is a water-soluble polysaccharide widely used to induce colitis in mice, with features resembling human ulcerative colitis [20]. The DSS-induced colitis model is very popular due to its rapidity, simplicity and controllability. Acute, chronic and relapsing models of intestinal inflammation can be achieved by modifying the concentration of DSS and the frequency of administration. The mechanism by which it induces colitis is unclear; however, this model does not require the activation of T and B cells, thus excluding adaptative immunity study, but favoring the study of the contribution of the innate immune system to the development of intestinal inflammation. In addition, the DSS colitogen likely induces epithelial apoptosis and consequent damages, and is, therefore, a useful model to study epithelial-dependent remodeling and inflammation in colitis [26]. In the present study, we demonstrate that IEC AMPK is an important player in the modulation of the sensitivity to DSS-induced colitis and regulation of mucosal repair after injury, suggesting the potential for targeting AMPK in the treatment of leaky gut disorders.

The role of AMPK in intestinal health has received increasing attention in recent years [10,27]. Studies using mouse models of colitis have showed a correlation between the decrease in intestinal AMPKα Thr172 phosphorylation (the active form of AMPK) and colitis development [15,17,28]. Interestingly, reduction in AMPKα Thr172 phosphorylation was also observed in the intestine of IBD patients and appeared to occur in both epithelial and *lamina propria* compartments [17], further highlighting the underlying connection between AMPK and intestinal barrier function. We recently reported that deletion of IEC AMPKα1 and AMPKα2 impaired intestinal barrier function in distal colon in homeostatic conditions [19]. Different studies have described how epithelial AMPK can control tight junction protein stabilization [29,30] and maintain epithelial barrier function following injury [12]. Consistently, altered formation of intestinal paracellular junction, as determined by immunostaining of E-cadherin and β-catenin markers, was reported in an ex vivo gut culture model established from E13.5-day fetal small intestine deleted for AMPKα1 [31]. Therefore, it is likely that higher permeability in the colon of IEC AMPK KO mice was a predisposing factor for the increased sensitivity to DSS-induced epithelial damage and colitis. In support, we observed a decrease in the gene expression levels of the tight junction complex proteins and a concomitant increase in pro-inflammatory gene expression in IEC AMPK KO compared to WT mice after DSS challenge. Accordingly, IEC AMPK KO mice presented increased intestinal permeability associated with severe histological damages. A previous report also showed that IEC AMPKα1 deletion in mice exacerbated colitis with higher disease activity index and histopathological score [11]. In this context, we assume that IEC AMPK is important for the maintenance of IEB function, particularly through the restitution of a functional epithelial lining after injury, indicating AMPK might be a key mediator involved in the etiology of IBD.

IEB is essential not only for maintaining mucosal homeostasis but also for wound repair. Mucosal repair of the intestinal barrier following injuries is a tightly coordinated response accomplished by a process termed epithelial restitution. Rapid resealing of the epithelium lining is crucial to restore the barrier function and a wound healing process dependent on epithelial cell proliferation, migration and differentiation is established to repair the damaged epithelium [32]. The ability of IECs to proliferate and regenerate is crucial for the restoration of the integrity of the mucosal barrier. Therefore, impaired IEC proliferation might explain why IEC-specific deletion of AMPK results in increased damage upon DSS treatment and associated altered wound healing response. In this context, the balance of epithelial proliferation and migration is also critical and migration of epithelial cells to the wound particularly contribute to mucosal wound closure [33]. It is possible that IEC AMPK participates to the migration of cells repopulating the wound as highlighted by recent studies showing the pivotal role of AMPK in the control of cell migration [34,35]. In line, loss of intestinal AMPKα1 impaired epithelial cell migration ex vivo and in vivo [11,31].

Our present study completes our knowledge on AMPK function on IEB integrity, showing that AMPK is not required for IEC adhesion or spreading, but is involved in IEC proliferation, epithelial barrier maturation and differentiation during the recovery from DSS challenge. This is consistent with previous studies showing that AMPK strengthens intestinal differentiation via epigenetically promoting Cdx2 expression [11]. Healing of the intestinal lining is also accompanied by the restitution of mucus-producing goblet cells, which perform important roles in tissue regeneration and maintenance of the intestinal epithelial barrier [36]. We observed that DSS-induced colitis in IEC AMPK KO mice led to a reduction in the number of colonic goblet cells, with a concomitant striking decrease in the expression of the major mucin-encoding gene *Muc2* responsible for mucus synthesis. Interestingly, deficient differentiation has been also demonstrated in an ex vivo gut culture model lacking AMPKα1 and was associated with a reduced prevalence of goblet cells in this epithelium developing model [31]. Previous studies have reported that cytokines, including IL-22 and IL-23a, produced by cells of the innate immune system enhance colonic restitution and accelerate restoration of goblet cells to promote mucosal healing during acute experimental colitis [37,38,39]. However, no changes were observed between IEC AMPK KO and WT mice for IL-22 and IL-23a gene expression levels in DSS-treated colon biopsies (unpublished data). Extensive loss of goblet cells is likely contributing to the delayed recovery following DSS-induced colitis. Indeed, mouse models with a reduced number of goblet cells display enhanced sensitivity to DSS-induced colonic injury, impaired proliferative capacity of IEC and inefficient colonic tissue repair [38,40,41]. Intestinal goblet cells facilitate mucosal protection and epithelial barrier repair by producing the mucus layer, which plays a crucial role as a physical barrier against bacterial translocation to the *lamina propria* [42], but also other molecules involved in tissue repair such as trefoil factors [43]. The mucus layer protects the intestinal epithelium from injury and pathologic conditions, such as IBD. Its alteration is associated with increased sensitivity to DSS-induced injury and inflammation, as observed in Muc2-deficient mice [44]. Furthermore, animals with a penetrable inner mucus layer develop spontaneous colitis. Goblet cell depletion is prominently observed in patients with ulcerative colitis [45]. Patients with active ulcerative colitis have a fully penetrable mucus, whereas patients in remission have a more variable profile, with some looking perfectly normal [46].

DSS administration is associated with alterations of the gut microbiota and deregulation of the immune response [47,48], analogous to modifications reported in human IBD [49,50]. Enteric bacteria are essential for the development of DSS-induced colitis, as evidenced by the prevention of inflammation by treatment with antibiotics [51,52]. The gut microbiota of DSS-treated mice has been characterized by a loss of bacterial diversity and changes in bacterial composition towards pro-inflammatory Gram-negative bacteria [53]. There is also increasing evidence that the initial gut microbiota composition influences the outcome of DSS-induced colitis in mice [54,55]. As IEC AMPK KO mice display a shift in the gut microbiota composition with higher levels of Clostridiales and Desulfovibrionales [19], these variations might also contribute to the exacerbated response to DSS-challenge and delayed mucosal repair. However, future studies examining this hypothesis are warranted.

Whereas IEB integrity loss is a well-recognized IBD contributing factor [1,24], the underlying molecular mechanisms of IEB failure as well as IEB reinforcing agents remain to be identified. Our work not only helps to understand the role of AMPK in the regulation of IEB in pathological conditions but also proposes the use of AMPK activators as IEB reinforcing agents in IBD. This is consistent with previous studies showing that treatment with the direct AMPK activator A-769662 concurrently to DSS challenge provided a protection against DSS-driven inflammation in mice [17]. Previous studies have also documented the ameliorative effects of the non-specific AMPK agonists, metformin and AICAR, in mouse models of colitis [15,16,17,18,56,57,58] and possibly a metformin protective effect in human IBD [59]. It has been suggested that these compounds act as central inhibitors of inflammatory and immune responses in experimental colitis via the inhibition of NF-κB activation, reduction of pro-inflammatory cytokines production and down-regulation of Th1 and Th17 immune responses [17,56,57,58]. These data suggest that targeting immune cells with AMPK activators represents a promising pharmacological tool against bowel inflammation. In support, salicylate, a direct AMPKβ1 activator, has been recently proven to be therapeutically effective in ameliorating inflammation in DSS-induced colitis by the activation of macrophage autophagy through an AMPK-dependent mechanism [60]. In the present study, to test the therapeutic relevance of IEC AMPK activation, we administrated metformin at the beginning of the recovery phase after withdrawal of DSS to limit the impact of metformin action on the inflammatory phase. We found that metformin enhanced colonic tissue repair through an IEC AMPK-independent mechanism. These data suggest that alternative pathways are able to bypass the requirement for IEC AMPK to regulate mucosal repair. In future studies, it will be interesting to determine the mechanisms by which metformin exerts these effects. Notwithstanding, it remains pertinent to address the efficacy of more selective direct AMPK activators on intestinal inflammation and mucosal repair to improve the management of IBD patients. Among the candidate compounds is the AMPK activator PXL-770, which entered Phase 2a clinical trials and was well tolerated [61].

## 5. Conclusions

Using AMPK WT and IEC AMPK KO mice as well as WT and AMPK KO Caco2 cells, we report that intestinal epithelial AMPK is involved in the maintenance of IEB integrity after injury. Disruption of intestinal epithelial AMPK increases the severity of DSS-induced intestinal epithelial injury and causes impaired restitution of goblet cells associated with altered proliferation and maturation of IEC, leading to a delayed recovery of IEB function. These results highlight the role of intestinal epithelial AMPK for the establishment of barrier resistance. However, the mechanism underlying the metformin-induced improvement of epithelial repair appears to be independent of IEC AMPK activation. Further understanding of the mechanisms by which metformin ameliorates colitis may give new insights to enhance IEB integrity.

## Figures and Tables

**Figure 1 cells-11-00590-f001:**
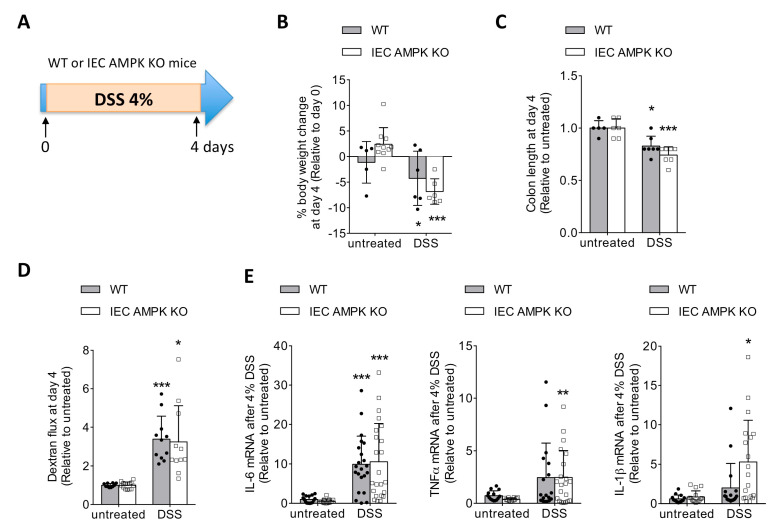
DSS-induced colonic injury and inflammation in WT and IEC AMPK KO mice. (**A**) Experimental timeline for 4% DSS-induced colitis in female WT and IEC AMPK KO mice. Mice were treated with 4% DSS in drinking water for four consecutive days. Changes in (**B**) body weight and (**C**) colon length in WT and IEC AMPK KO mice on day 4 of 4% DSS-induced colitis (*n* = 5–10 mice/group). (**D**) In vivo paracellular intestinal epithelial permeability of WT and IEC AMPK KO mice on day 4 during 4% DSS-induced colitis protocol. Dextran flux was determined by measuring the amount of 4 kDa TRITC-dextran in the plasma 4 h after gavage (*n* = 10–14 mice/group). (**E**) Expression of mRNA for Il-6, TNF-α and IL-1β in the colon from WT and IEC AMPK KO mice on day 4 (*n* = 16–23 mice/group). The results are pooled from four independent experiments. Data are expressed as means ± SD. Statistical analysis was performed by one-way ANOVA with the post hoc Bonferroni’s multiple comparisons test; * *p* < 0.05, ** *p* < 0.01 and *** *p* < 0.001 indicate a significant change relative to the untreated condition.

**Figure 2 cells-11-00590-f002:**
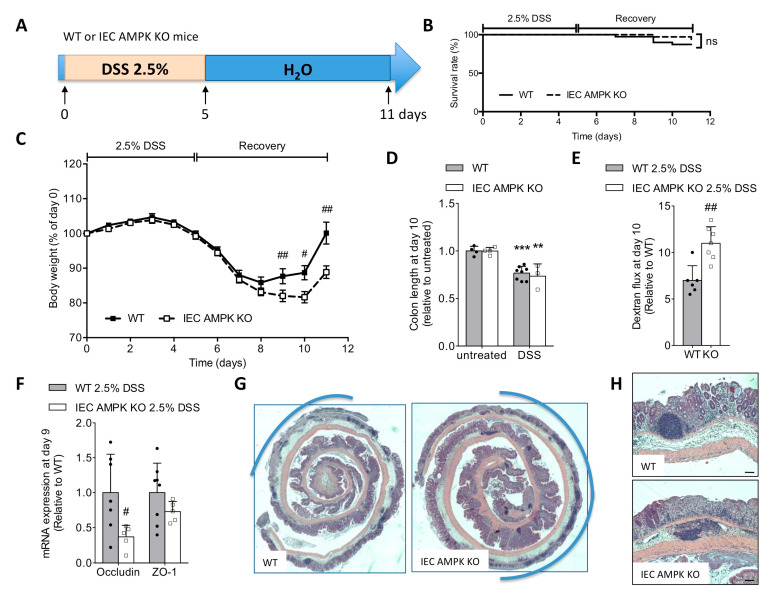
Absence of IEC AMPK delays intestinal epithelial recovery after DSS-induced injury. (**A**) Experimental timeline for 2.5% DSS-induced colitis in female WT and IEC AMPK KO mice. Mice were treated with 2.5% DSS in drinking water for five consecutive days, followed by six days of regular water. (**B**) Survival rate during the experimental period (*n* = 36–41 mice/group). The results are pooled from three independent experiments. (**C**) Evolution of body weight during the progression of colitis induced by 2.5% DSS in WT and IEC AMPK KO mice (*n* = 17–20 mice/group). The results are pooled from two independent experiments. (**D**) Colon length on day 10 following DSS challenge (*n* = 4–8 mice/group). (**E**) In vivo paracellular intestinal epithelial permeability of WT and IEC AMPK KO mice on day 9 during 2.5% DSS-induced colitis protocol. Dextran flux was determined by measuring the amount of 4 kDa TRITC-dextran in the plasma 4 h after gavage. Data are presented as mean dextran flux fold change relative to respective control mice (***n*** = 6–7 mice/group). (**F**) Expression of mRNA for tight junction proteins occludin and ZO-1 in the distal colon of WT and IEC AMPK KO mice on day 9 of the recovery phase (*n* = 5–8 mice/group). (**G**) Representative images of H&E-stained sections of Swiss-rolled colonic specimens (with proximal gut in the center and distal colon in the exterior) from IEC AMPK KO and WT mice on day 9. Pictures are representative of 10–13 mice/group. Ulceration is highlighted in blue. (**H**) Higher magnification images of colon sections. Scale bars = 50 µm. (**I**) Histological damage score (*n* = 10–13 mice/group) and quantification of percentage of colon ulceration (*n* = 5–7 mice/group). (**J**) Expression of mRNA and (**K**) plasma levels for pro-inflammatory cytokines Il-6, TNF-α and IL-1β in IEC AMPK KO and WT mice on day 9 (*n* = 5–7 mice). Data are expressed as means ± SD. Statistical analysis was performed by using a Student’s *t* test, non-parametric Mann–Whitney test, one-way ANOVA or two-way ANOVA with Bonferroni post hoc test; ** *p* < 0.01 and *** *p* < 0.001 indicate a significant difference relative to the untreated condition; # *p* < 0.05, ## *p* < 0.01 and ### *p* < 0.001 a significant difference between genotype; n.s., not significant.

**Figure 3 cells-11-00590-f003:**
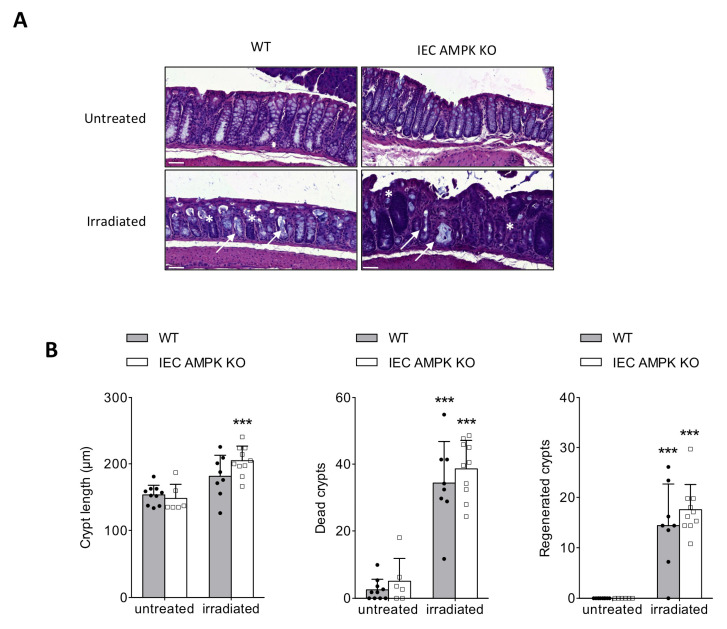
Altered cell proliferation but not adherence in AMPK-deficient Caco2 cells. (**A**) Representative H&E-stained colon from IEC AMPK KO and WT mice at 5 days after γ irradiation with 10 Gy. Pictures are representative of 6–10 mice/group. White arrow indicated dead crypts and * regenerated crypts. Scale bars = 50 µm. (**B**) Quantification of crypt length, and dead and regenerated crypts 5 days after irradiation (*n* = 6–10 mice/group). (**C**) Epithelial cell proliferation was assessed by immunostaining of Ki-67 during the recovery phase (day 8 to day 11) of DSS-induced colitis in IEC AMPK KO and WT mice. The graph shows the percentage of Ki-67-positive cells over total crypt length (*n* = 3–4 mice/group). The results were pooled from two triplicates. (**D**) Representative images of Ki67-stained colon sections following 2.5% DSS treatment on days 9 and 11. Pictures are representative of 3–4 mice/group. Scale bars = 50 µm. (**E**) WT and AMPK-deficient Caco2 cells proliferation was evaluated by xCELLigence real-time cell analysis (RTCA). Data are expressed as a cell index value corresponding to the proportion of the plate surface occupied by adherent cells. The results are representative of three replicates. (**F**) Proliferation was assessed with the use of the slope (h^−1^) of data collected with the xCELLigence software (*n* = 3 per genotype). (**G**) Adhesion of WT and AMPK-deficient Caco2 cells. The results are pooled from two independent experiments. Data are expressed as means ± SD. Statistical analysis was performed by using a Student’s *t* test, one-way ANOVA or two-way ANOVA with Bonferroni post hoc test; *** *p* < 0.001 indicates a significant difference relative to the untreated condition; ## *p* < 0.01 and ### *p* < 0.001 indicate a significant difference between genotype.

**Figure 4 cells-11-00590-f004:**
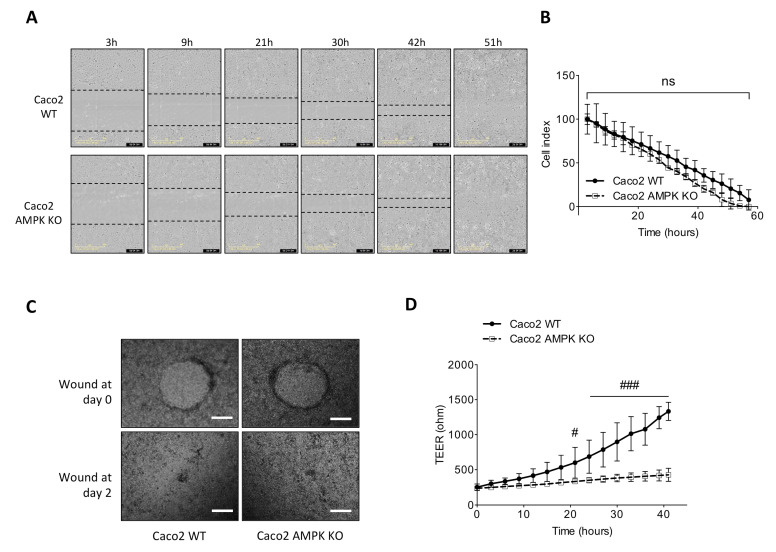
Altered cell spreading in AMPK-deficient Caco2 cells. (**A**) Representative acquisitions of monolayer of WT and AMPK-deficient Caco2 cells after wound scratch. Wound delimitations are indicated with dotted lines. Pictures are representative of two independent experiments performed in triplicate. (**B**) Quantification of relative wound width between WT and AMPK-deficient Caco2 cells was assessed by measuring the distance between the boundaries of the migrating cells over time. The results are representative of two independent experiments performed in triplicate. (**C**) Representative image of wound closure two days after injury on polarized confluent WT and AMPK-deficient Caco-2 cells. Pictures are representative of 20 wounds (4 wounds/transwell, 5 transwells per condition) from three independent experiments (scale bars = 200 μM). (**D**) Variations of TEER during wound closure upon damage on polarized confluent WT and AMPK-deficient Caco-2 cells. The results are pooled from 5 transwells per condition in three independent experiments. Data are expressed as means ± SD. Statistical analysis was performed by using two-way ANOVA with Bonferroni post hoc test; # *p* < 0.05 and ### *p* < 0.001 indicate a significant difference between genotype; n.s. not significant.

**Figure 5 cells-11-00590-f005:**
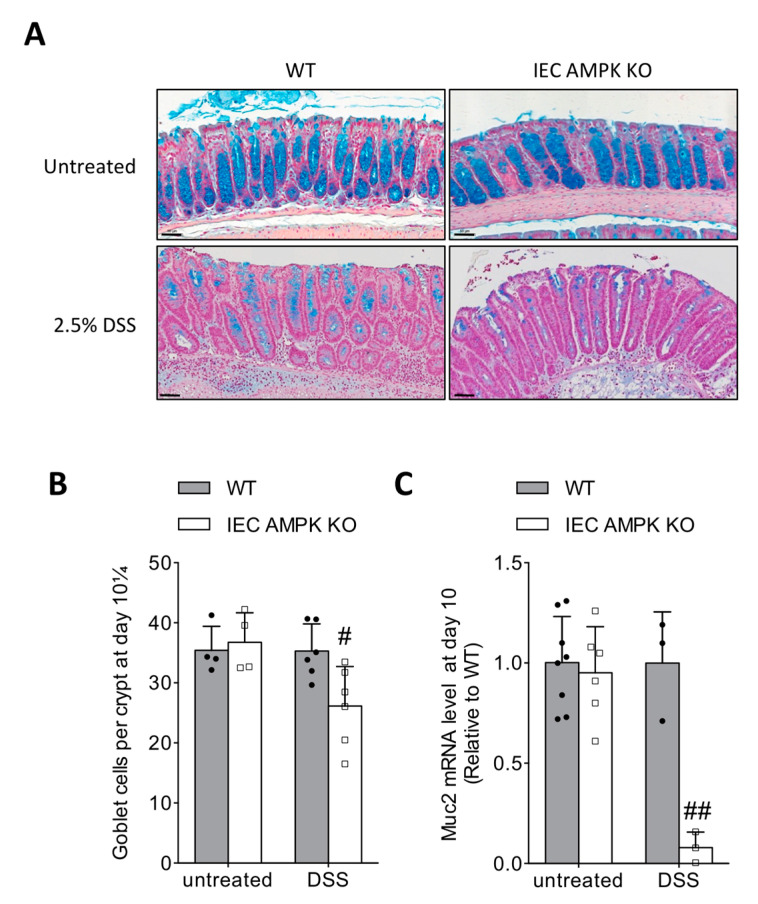
Altered goblet cell restitution after DSS-induced colitis in IEC AMPK KO cells. (**A**) High magnification of colon sections stained with Alcian Blue from WT and IEC AMPK KO mice untreated or after DSS-induced colitis on day 10. Pictures are representative of 4–6 mice/group. Scale bars = 50 µm. (**B**) Quantification of the number of goblet cells within the crypts of colons from WT and IEC AMPK KO mice untreated or after DSS challenge on day 10 (*n* = 4–6 mice/group). Goblet cell number was normalized by crypt depth (goblet cells/μm crypt depth) (**C**) Expression of mucin-encoding gene Muc2 (*n* = 3–8 mice/group). Data are expressed as means ± SD. Statistical analysis was performed by using a Student’s *t* test or one-way ANOVA with Bonferroni post hoc test; # *p* < 0.05 and ## *p* < 0.01 indicate a significant difference between genotype.

**Figure 6 cells-11-00590-f006:**
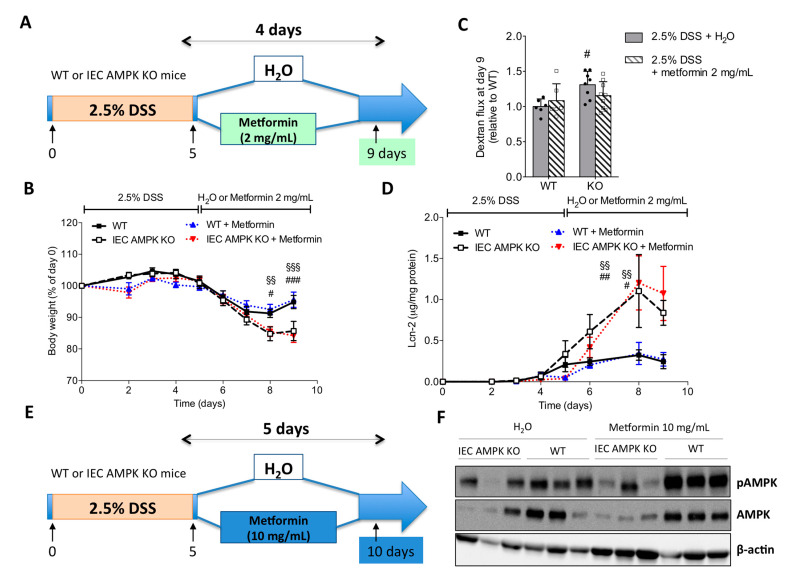
Effect of metformin treatment during the recovery phase on the recovery from DSS-induced colitis in WT and IEC AMPK KO mice. (**A**) Experimental timeline for mice treated with 2.5% DSS in drinking water for five consecutive days, followed by four days of metformin (2 mg/mL) in drinking water or regular water. (**B**) Changes in body weight (*n* = 6–9 mice/group). (**C**) In vivo intestinal epithelial permeability on day 9. Dextran flux was determined by measuring the amount of 4 kDa TRITC-dextran in the plasma 4 h after gavage. Data are presented as mean dextran flux fold change relative to respective control mice (*n* = 6–9 mice/group). (**D**) Lipocalin-2 (Lcn-2) levels in feces from WT and IEC AMPK KO mice during the course of DSS-induced colitis and treatment with metformin (2 mg/mL) in drinking water (*n* = 6–9 mice/group). (**E**) Experimental timeline for mice treated with 2.5% DSS in drinking water for five consecutive days, followed by five days of metformin (10 mg/mL) in drinking water or regular water. (**F**) Levels of total AMPKα and AMPKα-Thr172 phosphorylation in full-thickness colon biopsies from WT and IEC AMPK KO mice on metformin (10 mg/mL) in drinking water or regular water (*n* = 3 mice/group). (**G**) Quantification of AMPKα-Thr172 phosphorylation in Western blot from three replicates. (**H**) Changes in body weight (*n* = 6–9 mice/group). (**I**) In vivo intestinal epithelial permeability on day 10. Dextran flux was determined by measuring the amount of 4 kDa TRITC-dextran in the plasma 4 h after gavage. Data are presented as mean dextran flux fold change relative to respective control mice (*n* = 6–9 mice/group). (**J**) Representative images of H&E-stained colon sections on day 10 from IEC AMPK KO and WT DSS-treated mice administrated H_2_O or 10 mg/mL metformin during the recovery phase. Pictures are representative of 3–4 mice/group. Scale bars = 50 µm. (**K**) Histological damage score and quantification of percentage of ulceration related to total colon length in IEC AMPK KO and WT mice upon DSS-induced epithelial damage and metformin (10 mg/mL) treatment (*n* = 3–4 mice/group). Data are expressed as means ± SD. Statistical analysis was performed by one-way ANOVA or two-way ANOVA with Bonferroni post hoc test; * *p* < 0.05 and ** *p* < 0.01 indicate a significant difference relative to the untreated condition; # *p* < 0.05, ## *p* < 0.01 and ### *p* < 0.001 indicate a significant difference between genotype and §§ *p* < 0.01 and §§§ *p* < 0.001 indicate a significant difference between metformin-treated mice.

## Data Availability

The data presented in this study are available on request from the corresponding author.

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
