# Peer review of "Intestinal Epithelial AMPK Deficiency Causes Delayed Colonic Epithelial Repair in DSS-Induced Colitis"

_cells, 2022, doi:10.3390/cells11040590_

Round 1
Reviewer 1 Report
In the paper, entitled “Intestinal epithelial AMPK deficiency causes delayed colonic epithelial repair in DSS-induced colitis” by Olivier et al., the authors demonstrate a role for the AMPK expressed in intestinal epithelial cells (IEC) in the regulation of mucosal repair following DSS-induced colonic injury. In addition, they demonstrate a therapeutic potential of Metformin in this process that is independent of intestinal AMPK.
The paper is well written and the experiments have been nicely designed, rigorously conducted, with appropriate controls and sample sizes. However, the statistical analyses need to be revised but the conclusions may still be appropriately drawn based on the presented data.
The main concerns are the statistical analyses.
When comparing 2 groups, t- test could be appropriate assuming Gaussian distribution and similar standard deviation for both populations. Otherwise a non-parametric Mann Whitney test should be performed. More importantly, when comparing more than 2 groups, a one-way ANOVA or a Kruskal Wallis test should be performed depending of Gaussian distribution and standard deviations. A 2-way ANOVA is performed when studying more than 2 parameters and an ANOVA with repeated measures is preferred for kinetics analyses (as in Figure 2C, 3B …).
So the authors should review ALL their statistical analyses and be more explicit with the definition of their post-test comparisons. For example, on Figure 1C one-way ANOVA with Bonferroni post hoc test should be performed (and not 2-way ANOVA as indicated by the authors) and one should understand whether * corresponds to p<0.05 comparing DSS to untreated for a specific group or comparing WT to KO in one specific condition.
In addition, for clarity, data representations as scatter plots with mean and Standard Deviation is more informative than bar graphs with mean and SEM.
On Figure 2G, specifying the orientation of the colic swiss roles would help the reader (proximal gut in the center, distal colon in the exterior) in addition the legend does not explain the blue arrows, and an insert with a blowup of ulceration could be useful. Have the authors tried to quantify the ulcerations, inflammatory infiltrates and the different disease progression characteristics (loose feces versus diarrhea versus rectal bleeding)?
On Figure 3, representative pictures of Ki67 would be nice to illustrate quantification on Figure 3C. In addition, p9 line 336, the authors claim that their data indicate a delay in the regenerative process. The data indicate an alteration but no data specifically support the delay interpretation.
On Figure 3A, it seems that the number of Goblet cells is strongly different in the colon of non-treated WT mice versus non-treated IEC AMPK KO mice. The non-treated condition is missing in Figure 5A and 5B and is needed to correctly interpret the data. In addition, the quantification of Goblet cells displayed in Figure 5B is not clear, is it per crypt, per surface?
On Figure 4A, the images of the wound scratch experiment are really difficult to read on a printed version of the paper, the cells are invisible at the final scale. Could the authors provide more contrasted images, in addition, the doted red lines would be more visible as white or as black doted lines without any shadow.
The western blots presented on Figure 6F are puzzling. How do the authors explain that some AMPK and phospho-AMPK signals are detectable in the IEC KO groups? In their previous paper, no signal was detected in similar conditions.
Have the authors been able to quantify the water+metformin intake for the mice especially was the intake similar when water was with 2mg/ml versus 10mg/ml Metformin ?
Minor points
P12 the reference to the Figure 6 are completely wrong please correct lines 435 to 442
P12 line 446-448 remove sentence
P16 remove line 584 to 588
Author Response
Comments and Suggestions for Authors
In the paper, entitled “Intestinal epithelial AMPK deficiency causes delayed colonic epithelial repair in DSS-induced colitis” by Olivier et al., the authors demonstrate a role for the AMPK expressed in intestinal epithelial cells (IEC) in the regulation of mucosal repair following DSS-induced colonic injury. In addition, they demonstrate a therapeutic potential of Metformin in this process that is independent of intestinal AMPK.
The paper is well written and the experiments have been nicely designed, rigorously conducted, with appropriate controls and sample sizes. However, the statistical analyses need to be revised but the conclusions may still be appropriately drawn based on the presented data.
The main concerns are the statistical analyses.
When comparing 2 groups, t- test could be appropriate assuming Gaussian distribution and similar standard deviation for both populations. Otherwise a non-parametric Mann Whitney test should be performed. More importantly, when comparing more than 2 groups, a one-way ANOVA or a Kruskal Wallis test should be performed depending of Gaussian distribution and standard deviations. A 2-way ANOVA is performed when studying more than 2 parameters and an ANOVA with repeated measures is preferred for kinetics analyses (as in Figure 2C, 3B …).
So the authors should review ALL their statistical analyses and be more explicit with the definition of their post-test comparisons. For example, on Figure 1C one-way ANOVA with Bonferroni post hoc test should be performed (and not 2-way ANOVA as indicated by the authors) and one should understand whether * corresponds to p<0.05 comparing DSS to untreated for a specific group or comparing WT to KO in one specific condition.
In addition, for clarity, data representations as scatter plots with mean and Standard Deviation is more informative than bar graphs with mean and SEM.
We highly appreciate this reviewer’s very positive comments on our study. We thank the reviewer for the very competent comments and suggestions, and the opportunity to improve our manuscript. In the revised submission, we reviewed all statistical analyses and corrected the figure legends accordingly. We also modified data representations as scatter plots with mean and Standard Deviation. We believe that all these changes have made the manuscript better.
On Figure 2G, specifying the orientation of the colic swiss roles would help the reader (proximal gut in the center, distal colon in the exterior) in addition the legend does not explain the blue arrows, and an insert with a blowup of ulceration could be useful. Have the authors tried to quantify the ulcerations, inflammatory infiltrates and the different disease progression characteristics (loose feces versus diarrhea versus rectal bleeding)?
On Figure 2G, we added information on the orientation of the swiss rolls and provided an evaluation of percentage of colon ulceration and inflammatory infiltrates by scoring colonic histological damage (Figure 2I). The effect of DSS treatment on the time course of disease progression was monitored by the measure of weight loss (Figure 2C). We also measured the appearance of loose feces, diarrhea and rectal bleeding but evaluation of these later parameters was not sufficient to reveal obvious difference in the signs of illness and severity between WT and IEC AMPK KO mice.
On Figure 3, representative pictures of Ki67 would be nice to illustrate quantification on Figure 3C. In addition, p9 line 336, the authors claim that their data indicate a delay in the regenerative process. The data indicate an alteration but no data specifically support the delay interpretation.
On Figure 3, representative pictures for Ki67 have been added and line 336, page 9 has been modified according to the reviewer’s suggestion.
On Figure 3A, it seems that the number of Goblet cells is strongly different in the colon of non-treated WT mice versus non-treated IEC AMPK KO mice. The non-treated condition is missing in Figure 5A and 5B and is needed to correctly interpret the data. In addition, the quantification of Goblet cells displayed in Figure 5B is not clear, is it per crypt, per surface?
We quantified the number of Globet cells in non-treated WT and IEC AMPK KO colon and no noticeable differences were observed. These data for non-treated condition have been added in Figures 5A and B. These results are supported by similar Muc2 gene expression in non-treated WT and IEC AMPK KO mice (1.00 ± 0.08 vs 0.95 ± 0.09 relative to WT; not significant). The number of Globlet cells has been determined per crypt and was normalized by crypt depth (goblet cells/μm crypt depth).
On Figure 4A, the images of the wound scratch experiment are really difficult to read on a printed version of the paper, the cells are invisible at the final scale. Could the authors provide more contrasted images, in addition, the doted red lines would be more visible as white or as black doted lines without any shadow.
On Figure 4A, the images of wound scratch have been modified to increase contrast and have a better visibility of these data.
The western blots presented on Figure 6F are puzzling. How do the authors explain that some AMPK and phospho-AMPK signals are detectable in the IEC KO groups? In their previous paper, no signal was detected in similar conditions.
For Western blot analysis, we used in the present study full-thickness colon tissue samples. The presence of some AMPK and phospho-AMPK signal in IEC KO groups likely derives from non-IEC cells present in smooth muscle and lamina propria. AMPK expression in non-IEC cells is not expected to be affected by expression of Cre under the control of the villin promoter to drive Cre expression. In previous paper (Mol Metab. 2021 Feb 4;47:101183), Western blots were realized with protein extracts obtained from isolated IECs (after elimination of non-IEC cells).
Have the authors been able to quantify the water+metformin intake for the mice especially was the intake similar when water was with 2mg/ml versus 10mg/ml Metformin ?
We have quantified the water+metformin intake during the time course of DSS treatment and the recovery phase. We found no difference in the daily intake per cages treated with water or water + 2 mg/ml metformin during the progression of the disease. Unfortunately, water+metformin intake has not been monitored in the water + 10 mg/ml Metformin experimental group. However, we checked the effect of 10 mg/ml Metformin administration on the phosphorylation of AMPK in the colon. As reported in Figures 6F and G, AMPK Thr172 phsophorylation was substantially increased upon metformin treatment in WT but not IEC AMPK KO mice.
Minor points
P12 the reference to the Figure 6 are completely wrong please correct lines 435 to 442
We thank the reviewer for his/her carefully reading our manuscript. The reference to Figure 6 has been corrected.
P12 line 446-448 remove sentence
This sentence has been removed.
P16 remove line 584 to 588
These lines have been removed.

Reviewer 2 Report
In the present manuscript, Olivier et al. have investigated the role of intestinal epithelial AMPK in response to DSS-induced acute colitis. Using a unique intestinal epithelial cell-specific AMPK KO mice, they mainly reported that deletion of the kinase delays tissue repair after acute colitis. By exploring some of the putative underlying mechanism(s), the authors showed that AMPK apparently plays a role in re-epithelization of the intestinal mucosa.
Altogether, the manuscript is interesting and well-written, the experimental approaches used are solid and the data generally convincing. The interpretation and discussion are well balanced. I only have few minor comments listed below.
Page 2, line 51: I'd suggest replacing "fuel-sensitive" by "nutrient-sensitive"
Page 4, line 157: typo "descibed"
Page 8, Figure 2I: did the authors performed outlier test ? the data for IL1b level in WT mice seeems to have a much larger spread than the other conditions
Page 9, line 342-344: this sentence is unclear; please rephrase
Page 9, Figure 3F: the x-axis legend is missing (time ?)
Page 11, Figure 5B: stat missing ? if borderline significant, please indicate p value on the graph
Page 12-13, Figure 6: the figure organization and panel numbers are wrong in the text description; WB for pAMPK and AMPK in panel F look not very homogenous (AMPK expression in some KO), please check if correct (shift ?). p/T AMPK quantification may eventually be added at the bottom of the representative blots.
Page 12, line 446-448: should be removed
Discussion: few lines may be added on the pros and cons of DSS exposure as acute colitis model
Author Response
Comments and Suggestions for Authors
In the present manuscript, Olivier et al. have investigated the role of intestinal epithelial AMPK in response to DSS-induced acute colitis. Using a unique intestinal epithelial cell-specific AMPK KO mice, they mainly reported that deletion of the kinase delays tissue repair after acute colitis. By exploring some of the putative underlying mechanism(s), the authors showed that AMPK apparently plays a role in re-epithelization of the intestinal mucosa.
Altogether, the manuscript is interesting and well-written, the experimental approaches used are solid and the data generally convincing. The interpretation and discussion are well balanced. I only have few minor comments listed below.
Page 2, line 51: I'd suggest replacing "fuel-sensitive" by "nutrient-sensitive"
We thank the reviewer for his/her comments and propositions. We think this contributes largely to improve our manuscript. Fuel-sensitive has been changed to nutrient-sensitive as suggested. The manuscript has been changed accordingly..
Page 4, line 157: typo "descibed"
This typo has been corrected.
Page 8, Figure 2I: did the authors performed outlier test ? the data for IL1b level in WT mice seems to have a much larger spread than the other conditions
We thank the reviewer for pointing out this important point. After performing outlier test, we removed non-relevant points and re-analyzed the data. We found that circulating IL-1b levels were significantly higher in IEC AMPK KO compared to WT mice, indicating that loss of IEC AMPK could contribute to the development of a low grade basal chronic inflammation during DSS-induced colitis.
Page 9, line 342-344: this sentence is unclear; please rephrase
This sentence has been changed in the revised manuscript.
Page 9, Figure 3F: the x-axis legend is missing (time ?)
X-axis-legend has been added.
Page 11, Figure 5B: stat missing ? if borderline significant, please indicate p value on the graph
We thank the reviewer for this remark.. We realized that the statistical analysis was missing in this figure and it has now been corrected accordingly.
Page 12-13, Figure 6: the figure organization and panel numbers are wrong in the text description; WB for pAMPK and AMPK in panel F look not very homogenous (AMPK expression in some KO), please check if correct (shift ?). p/T AMPK quantification may eventually be added at the bottom of the representative blots.
We corrected the panel numbers for this figure and the text description. We agree that the Western blot signals for pAMPK/ AMPK are not very homogenous. This is caused by the use in the present study of full-thickness colon tissue sample extracts. The presence of some AMPK and phospho-AMPK signal in IEC KO groups likely derives from non-IEC cells present in smooth muscle and lamina propria. We provided a quantification of pAMPK and showed a significant increase in colonic AMPK phosphorylation in response to metformin treatment in WT but not IEC AMPK KO mice (Figure 6G).
Page 12, line 446-448: should be removed
This sentence have been removed. We thank the reviewer for his/her alertness.
Discussion: few lines may be added on the pros and cons of DSS exposure as acute colitis model
As requested by this reviewer, we have now expanded in Discussion section about the pros and cons of DSS experimental model.

Reviewer 3 Report
In this manuscript, the author investigated the role of intestinal epithelial cell AMPK in colitis recovery. IEC AMPK delayed intestinal injury repair after DSS exposure and was associated with a slower re-epithelization of the intestinal mucosa and altered barrier function. IEC AMPK KO mice also displayed a lower goblet cell counts with concomitant decreased Muc2 gene expression. Metformin treatment during the recovery phase reduced the severity of DSS-induced colitis through improvement in intestinal repair capacity in both WT and IEC AMPK KO mice. The data is clearly presented and the manuscript is well written and easy to follow, but several questions need to be addressed.
- The author needs to check figure order and caption in Figure 2. The figure order in Figure 2 doesn't match the manuscript.
- In Figure 2, The authors saw WT mice exhibited accelerated body weight gain and gradually recover faster than IEC AMPK KO mice at the end of the recovery phase, however, colon length at day 10 didn't display too much difference between WT and KO mice. Did the author evaluate other colitis parameters like loose feces, rectal bleeding, or diarrhea between WT and KO mice during the recovery?
- Figure 2G and 2H, The author needs to use high magnification to display a more precise colon structure and to display the more massive inflammatory cell infiltration in KO mice. The author also needs to do the quantifications.
- The figure legend of Figure 2H said "(H) Expression of mRNA and (I) serum levels for pro-inflammatory cytokines Il-6, TNF-a and IL-1b in IEC AMPK KO and WT mice at day 11. ", however, the author claimed the mRNA level for pro-inflammatory cytokines were detected at day 9 of the recovery phase in the manuscript. The author needs to make it clear to the reader.
- In Figure 2C, according to the weight change, mice started to recover from DSS at day9, and differences between WT and KO at day10 and day11 are bigger than day9. Why did the author detect the mRNA level of pro-inflammatory cytokines on day9 of the recovery phase?
- Figure 3C, The result of Ki67 staining is inconclusive. First, the authors need to clarify how Ki67 was stained and quantified in the intestinal epithelium in the material and method section. Did the author co-stain Ki67 with the epithelium cell marker and how Ki67 was quantified? Second, although %Ki67 cells in the WT on the day 8,9,10 and11 were significantly higher than the day8 of KO, %Ki67 increased in KO at day9, 10, moreover, %Ki67 at day 10 of KO was higher than that in WT. The author needs to explain and discuss this phenotype and draw a conclusion cautiously. And also, representative pictures of Ki67 staining need to be shown.
- Figure3F. The author needs to clarify how the adherence ability of the Caco2 cell was measured.
- Figure5B. Is the globlet cell number significantly different between WT and KO?
- "The decreased quantity of globlet cells was associated with a strong reduction of the goblet cell marker mucin (Muc)-2 expression in the colon of IEC AMPK KO as compared to WT mice (Figure 5B)" Here "Figure 5B" should be "Figure 5C"
- The author needs to check the figure order in Figure 6
- Figure6A-D. The author didn't see effects on body weight changes nor improvement of the intestinal epithelial barrier function at day 9, nor reduction in intestinal inflammation when treated mice with 2mg/ml metformin. Did the author see AMPK T172 increase in WT mice after 2mg/ml metformin treatment?
- The conclusion of Figure 6 is unclear. The author claim that the high dose of metformin reduced the severity of DSS-induced colitis and reduced colon ulceration after damage, however, no relevant effects were noticed on body weight regain during the recovery phase in IEC AMPK KO and WT mice. Does that mean the reduction of colon ulceration has nothing to do with weight regain and recovery? The high dose of metformin reduced Dextran flux in WT but not in KO. This means the regulation of gut leakiness by metformin is dependent on IEC AMPK, however, same as the reduction of colon ulceration, it didn't help WT mice to regain more weight during the recovery phase. The author needs to address these phenotypes.
- Line 446-448 in the manuscript seems to be unrelated to any figure or conclusion, please check.
Author Response
Comments and Suggestions for Authors
In this manuscript, the author investigated the role of intestinal epithelial cell AMPK in colitis recovery. IEC AMPK delayed intestinal injury repair after DSS exposure and was associated with a slower re-epithelization of the intestinal mucosa and altered barrier function. IEC AMPK KO mice also displayed a lower goblet cell counts with concomitant decreased Muc2 gene expression. Metformin treatment during the recovery phase reduced the severity of DSS-induced colitis through improvement in intestinal repair capacity in both WT and IEC AMPK KO mice. The data is clearly presented and the manuscript is well written and easy to follow, but several questions need to be addressed.
- The author needs to check figure order and caption in Figure 2. The figure order in Figure 2 doesn't match the manuscript.
We thank the reviewer for his/her carefully reading of our manuscript. Figure 2 order and captions were carefully revised and reference in the manuscript corrected.
- In Figure 2, The authors saw WT mice exhibited accelerated body weight gain and gradually recover faster than IEC AMPK KO mice at the end of the recovery phase, however, colon length at day 10 didn't display too much difference between WT and KO mice. Did the author evaluate other colitis parameters like loose feces, rectal bleeding, or diarrhea between WT and KO mice during the recovery?
We thank the reviewer for his/her comment. Indeed, we were generally puzzled by the lack of difference in colon length at day 10. Of note, several studies indicated that the shortening of average colon lengths as more severe inflammation was present and severity of the colitis increases. By measuring colon length at day 10, a condition with milder inflammation, we assume that the impact on colon length is minored.
The effect of DSS treatment on disease progression was mostly monitored by the changes in body weight (Figure 2C). We also measured the appearance of loose feces, diarrhea and rectal bleeding but evaluation of these parameters was not sufficient to reveal obvious difference in the signs of illness and severity during the recovery phase between WT and IEC AMPK KO mice.
Figure 2G and 2H, The author needs to use high magnification to display a more precise colon structure and to display the more massive inflammatory cell infiltration in KO mice. The author also needs to do the quantifications.
We thank the reviewer for raising this interesting point. As suggested, we added high magnification pictures (Figure 2H) to illustrate inflammatory cell infiltration and provided an histological damage scoring based on inflammatory cell infiltration, muscle thickening, and loss of goblet cells (Figure 2I).
- The figure legend of Figure 2H said "(H) Expression of mRNA and (I) serum levels for pro-inflammatory cytokines Il-6, TNF-a and IL-1b in IEC AMPK KO and WT mice at day ", however, the author claimed the mRNA level for pro-inflammatory cytokines were detected at day 9 of the recovery phase in the manuscript. The author needs to make it clear to the reader.
We thank the reviewer for his/her alertness. Indeed, there was a mistake in the figure legend. Pro-inflammatory cytokines were measured at day 9.. This has been corrected in the revised manuscript.
- In Figure 2C, according to the weight change, mice started to recover from DSS at day9, and differences between WT and KO at day10 and day11 are bigger than day9. Why did the author detect the mRNA level of pro-inflammatory cytokines on day9 of the recovery phase?
Follow-up of body weight loss and body weight re-gain was reported in Figure 2C and we agree that a bigger difference in body weight recovery is observed at day 10 and even more pronounced at day 1 between WT and IEC AMPK KO mice. However, we did not carry out our functional analyses at all time points to evaluate the impact of IEC AMPK deletion on progresion of DSS-induced colitis. We decided to perform most of our at day 9 when IEC AMPK KO mice started to show significant difference in body weight re-gain and intestinal permeability. In addition, we also choose to monitor mRNA level of pro-inflammatory cytokines at day 9 because the inflammatory profile of DSS-treated mice appears to peak at day 8 as shown by the measure of Lcn-2 levels during the progression of the disease (Figure 6D).
- Figure 3C, The result of Ki67 staining is inconclusive. First, the authors need to clarify how Ki67 was stained and quantified in the intestinal epithelium in the material and method section. Did the author co-stain Ki67 with the epithelium cell marker and how Ki67 was quantified? Second, although %Ki67 cells in the WT on the day 8,9,10 and11 were significantly higher than the day8 of KO, %Ki67 increased in KO at day9, 10, moreover, %Ki67 at day 10 of KO was higher than that in WT. The author needs to explain and discuss this phenotype and draw a conclusion cautiously. And also, representative pictures of Ki67 staining need to be shown.
We thank the reviewer for his/her remarks and raising questions on the interpretation of Ki67 staining data. They greatly help us to improve the quality of our manuscript. In the Material and Methods section we added necessary details on Ki67 staining procedure. Ki67 positive cells were counted over total crypt length. Representative pictures of Ki67 staining were also added (Figure 3D). The observation of marked attenuation of the proliferative response to DSS-induced injury on day 8 in IEC AMPK KO mice compared to WT mice (Figure 3C), indicating an alteration in the regenerative process. This could be interpreted as a delay rather than an intrinsic defect in the capacity of AMPK KO ISCs to promote epithelium regeneration. Indeed, Ki67 staining at days 10 and 11 indicates that crypt cells lacking AMPK maintains a proliferative potential. This stimulated proliferation in crypts of epithelial AMPK deficient gut could be due to immune cells and / or mesenchymal cells underneath crypts, which secrete growth factors and cytokines crucial for tissue regeneration and crypt cell proliferation. Lastly, colonic epithelium repair is closely related to ISCs proliferation, migration and differentiation and whether loss of AMPK in ISCs contributes to the altered regeneration phase in IEC AMPK KO mice DSS-induced colitis remains unclear and need to be further investigated in the future using mice with ISC-specific deletion of AMPK.
- The author needs to clarify how the adherence ability of the Caco2 cell was measured.
Material and Method section has been completed regarding the measurement of adherence of Caco2 cells.
- Is the globlet cell number significantly different between WT and KO?
We thank the reviewer for his/her comment. We realized that the statistical analysis was missing in this figure and it has now been corrected accordingly. In addition, we also quantified the number of globlet cells in non-treated WT and IEC AMPK KO and we report no noticeable differences in this control condition.
- "The decreased quantity of globlet cells was associated with a strong reduction of the goblet cell marker mucin (Muc)-2 expression in the colon of IEC AMPK KO as compared to WT mice (Figure 5B)" Here "Figure 5B" should be "Figure 5C"
This error has been corrected.
- The author needs to check the figure order in Figure 6
Order of all figures has been checked and corrected.
- Figure6A-D. The author didn't see effects on body weight changes nor improvement of the intestinal epithelial barrier function at day 9, nor reduction in intestinal inflammation when treated mice with 2mg/ml metformin. Did the author see AMPK T172 increase in WT mice after 2mg/ml metformin treatment?
We admit that the lack of effect of 2 mg/ml metformin on improvement of intestinal barrier function was disappointing. Although we checked that water and water+ 2 mg/ml metformin intake was similar between the group of mice administrated with DSS, we did not examined the impact on AMPK Thr172 phosphorylation. We then choose to use a higher dose at 10 mg/ml metformin, which was efficient to enhance AMPK phosphorylation in the colon (Figures 6F and G).
- The conclusion of Figure 6 is unclear. The author claim that the high dose of metformin reduced the severity of DSS-induced colitis and reduced colon ulceration after damage, however, no relevant effects were noticed on body weight regain during the recovery phase in IEC AMPK KO and WT mice. Does that mean the reduction of colon ulceration has nothing to do with weight regain and recovery? The high dose of metformin reduced Dextran flux in WT but not in KO. This means the regulation of gut leakiness by metformin is dependent on IEC AMPK, however, same as the reduction of colon ulceration, it didn't help WT mice to regain more weight during the recovery phase. The author needs to address these phenotypes.
We thank the reviewer for his/her comments on our results on metformin treatment. We observed that metformin treatment was not sufficient to influence body weight although colon ulceration was improved. This result is puzzling but when body weight variations are related to colitis development and therefore colon ulceration, it is not exclusively dependent on colonic remodeling, and these two features of colitis are not occurring simultaneously in the mice. DSS-treated C57BL/6 mice could initially (around day 4) present a slight increase in body weight whereas their colons are already damaged, and mice continued to lose body weight after DSS removal when repair processes are taking place. Faster weight gain cannot be ruled out in metformin treated animals after day 10. But additionnaly, we were surprised to observe an absence of metformin impact on global intestinal permeability, and have therefore to consider the possibility that metformin regulates the small intestine, where the absorption of nutrients occurs and for which we have demonstrated an absence of AMPK involvement (AMPK regulation of permeability was observed only in the distal colon). Of note, effect of metformin on intestinal permeability was not significant (Figure 6I), therefore, it is difficult to conclude that metformin is dependent on IEC AMPK. We have revised the results section to make our conclusions more clear for the readers.
- Line 446-448 in the manuscript seems to be unrelated to any figure or conclusion, please check.
This sentence has been deleted.

Round 2
Reviewer 3 Report
The manuscript was well revised and questions were well addressed. It's OK to publish in the journal.
Author Response
We thank the reviewer for the positive evaluation.